# Gluten-Free Bread Enriched with Artichoke Leaf Extract In Vitro Exerted Antioxidant and Anti-Inflammatory Properties

**DOI:** 10.3390/antiox12040845

**Published:** 2023-04-01

**Authors:** Mirco Vacca, Daniela Pinto, Alessandro Annunziato, Arianna Ressa, Maria Calasso, Erica Pontonio, Giuseppe Celano, Maria De Angelis

**Affiliations:** 1Department of Soil, Plant and Food Science (DiSSPA), University of Bari Aldo Moro, 70126 Bari, Italy; mirco.vacca@uniba.it (M.V.); alessandro.annunziato@uniba.it (A.A.); arianna.ressa@uniba.it (A.R.); maria.calasso@uniba.it (M.C.); maria.deangelis@uniba.it (M.D.A.); 2Human Microbiome Advanced Project-HMPA, Giuliani SpA, 20129 Milan, Italy; dpinto@giulianipharma.com

**Keywords:** artichoke, chlorogenic acid, antioxidants, gluten-free, celiac disease, nutrigenomics, adjuvant therapy

## Abstract

Due to its high nutritional value and broad beneficial effects, the artichoke plant (*Cynara cardunculus* L.) is an excellent healthy food candidate. Additionally, the artichoke by-products are usually discarded even though they still contain a huge concentration of dietary fibers, phenolic acids, and other micronutrients. The present work aimed to characterize a laboratory-made gluten-free bread (B) using rice flour supplemented with a powdered extract from artichoke leaves (AEs). The AE, accounting for the 5% of titratable chlorogenic acid, was added to the experimental gluten-free bread. Accounting for different combinations, four different bread batches were prepared. To evaluate the differences, a gluten-free type-II sourdough (tII-SD) was added in two doughs (SB and SB-AE), while the related controls (YB and YB-AE) did not contain the tII-SD. Profiling the digested bread samples, SB showed the lowest glycemic index, while SB-AE showed the highest antioxidant properties. The digested samples were also fermented in fecal batches containing viable cells from fecal microbiota samples obtained from healthy donors. Based on plate counts, no clear tendencies emerged concerning the analyzed microbial patterns; by contrast, when profiling volatile organic compounds, significant differences were observed in SB-AE, exhibiting the highest scores of hydrocinnamic and cyclohexanecarboxylic acids. The fecal fermented supernatants were recovered and assayed for healthy properties on human keratinocyte cell lines against oxidative stress and for effectiveness in modulating the expression of proinflammatory cytokines in Caco-2 cells. While the first assay emphasized the contribution of AE to protect against stressor agents, the latter enlightened how the combination of SB with AE decreased the cellular TNF-α and IL1-β expression. In conclusion, this preliminary study suggests that the combination of AE with sourdough biotechnology could be a promising tool to increase the nutritional and healthy features of gluten-free bread.

## 1. Introduction

Wheat-based bakery products represent staple foods for many dietary regimens. These foods traditionally contain gluten that, based on its unique network-forming properties, is essential to obtain an optimal structure of leavened derivatives [1]. However, gluten-containing products are not suitable for all consumers. A recent meta-analysis determined the prevalence of celiac disease (CeD) at 0.7% or 1.4% of the population based on biopsy-proven screening or serologic tests, respectively [2]. Moreover, the occurrence of non-celiac gluten-related disorders (GrDs), e.g., non-celiac gluten sensitivity, wheat-dependent food allergy, respiratory allergy, dermatitis herpetiformis, gluten ataxia, and gluten-sensitive irritable bowel syndrome (IBS) [3], are leading to a growing demand of gluten-free (GF) products [4].

A life-long adherence to strict GF diets (GFDs) can determine a low daily consumption of prebiotics, e.g., fructans and arabinoxylans, that naturally occur in wheat [5,6]. Based on this assumption, previous studies investigated the GFD effect on gut microbiota and observed how it negatively affected the gut microbial communities of both GrD patients [7] and healthy subjects [8]. In a recent study [9], we discussed how the GFD was not sufficient to reverse the CeD pathophysiological status featured by elevated nitric oxide metabolites in serum [10] and urine [11]. In fact, considering the high oxidative stress status as a CeD biomarker [12,13,14,15], it seemed not to be exclusively related to the CeD diagnosis because it was found almost unchanged during the disease follow-ups despite the GFD [16]. Therefore, an antioxidant administration should be considered a valuable strategy to avoid or, at least, reduce cellular oxidation in CeD.

Many foods are natural sources of flavonoids and polyphenols [17], such as fruits and vegetables, including grapes/wine [18], berries [19], citrus fruits [20], pomegranates [21], and spices (e.g., oregano, thyme, rosemary, sage) [22], suggesting that the diet can be considered as a tool to provide antioxidants to consumers without resorting to drug treatments. Even the artichoke plant (*Cynara cardunculus* L.) is an excellent candidate for being considered a healthy food due to having antioxidant properties. Originating from the Mediterranean Basin, artichoke is widely appreciated for the presence of bioactive substances such as fibers, vitamins (A, B1, and C), and minerals, and it is rich in natural antioxidants such as chlorogenic acid, cynarin, 7-O-rutinoside luteolin, and 7-O-glucoside luteolin [23]. Unfortunately, the rate of artichoke-derived waste (i.e., roots, stems, bracts, and external leaves) generated during industrial processes is around 70–85% of the plant weight [24]. Because artichoke by-products are still enriched in beneficial phenols with recognized antioxidant, antiradical, anticarcinogenic, and antiapoptotic properties, the recent literature supports the application of various methodologies involving phenolic molecule recovery [25,26,27].

In CeD, another field of interest is the assessment of the nutritional balance caused by GFDs. The study carried out by Melini and Melini suggests that a considerable number of GF food products feature high energy and fats [28]. Similarly, a recent meta-analysis revealed that GF bread products are usually characterized by too high glycemic index values [29]. In CeD patients adhering to GFDs, the exposure to high caloric intakes is also supported by studies revealing an unbalanced glucose metabolism [30,31,32]. To overcome this problem, studies on sourdough expanded the knowledge about advantages determined by its application to leavened products. Differences between sourdough products were recently described by De Vuyst et al. [33], defining type-I, type-II, and type-III sourdough (tI-, tII-, and tIII-SD, respectively) as flour–water mixture spontaneously fermented based on a daily back-slopping in tI-SD and doughs obtained from starter-culture-initiated fermentation processes in tII- and tIII-SD. Without considering type-specific differences, SD biotechnology has demonstrated the possibility to enhance the nutritional value of leavened foods and the opportunity to label these products as ones with “lower glycemic index” [34,35,36]. Hence, in doughs, microbial fermentation can represent a suitable tool to reduce high energy and glycemic indices and allow researchers to obtain suitable products for CeD patients.

Therefore, the aim of this work was the development of a conventional food product with improved features in supporting the well-being of GrD and CeD patients. Thus, this work examined the antioxidant and anti-inflammatory properties of GF bread supplemented with artichoke leaf powder extract. Additionally, with the purpose of exploiting sourdough for reducing the glycemic index of derivatives, this work evaluated the features derived from the combination of sourdough and artichoke extract. The antioxidant and anti-inflammatory properties were investigated following an integrated multi-step approach based on in vitro assays, fecal microbiota contribution in digestion, and the resulting ex vivo effectiveness of digested bread samples in human cells.

## 2. Materials and Methods

### 2.1. Sourdough Preparation and Selection

Type-II sourdough (tII-SD) with a dough yield (DY, dough weight × 100/flour weight) of 200 was prepared using commercial rice flour (Easyglut, Pedon SpA; Colceresa, Italy) and a single-strain inoculum of *Leuconostoc pseudomesenteroides* DSM 20193. Six batches of tII-SDs were prepared by combining different incubation temperatures at 20, 25, and 30 °C with the starter cell density at 6 or 7 log CFU/mL. After 24 h of incubation under stirring conditions (150× *g*), pH values and the lactic acid bacteria (LAB) density was collected (Appendix A) to evaluate the best fermentation conditions.

The pH was determined by an ultrabasic ub-10 pH meter (Denver Instrument Company; Arvada, CO, USA) equipped with a food penetration probe. To determine the LAB cell density, 5 g of each tII-SD was suspended in 45 mL of a sterile sodium chloride solution (0.9 g/L) and homogenized in a Bag Mixer 400 P (Interscience, St Nom, France) at room temperature. Serial 10-fold dilutions were then plated with De Man, Rogosa, and Sharpe agar media (MRS agar; Oxoid, Basingstoke, Hampshire, UK) modified with the addition of 1% (*w*/*w*) maltose and 5% (*w*/*w*) yeast extract and adjusting the pH to 5.6 value. The plated LAB counts were incubated for 48 h at 30 °C.

Thus, the selected tII-SD, which was that obtained after 24 h of incubation at 25 °C and 6 log CFU/mL of starter inoculum, was further characterized for the total titratable acidity (TTA) [37] and organic acid concentration by using an ACS Acetic Acid Assay Kit and D-/L-Lactic Acid (D-/L-Lactate) Kit [38]. Both kits were purchased from Megazyme (Megazyme International Ireland Limited, Wicklow, Ireland).

### 2.2. Artichoke Leaf Extract

The water-soluble powder extract of artichoke leaves (AEs) characterized by 5% of titratable chlorogenic acid stabilized in maltodextrins was provided by Farmalabor S.r.l. (Canosa di Puglia, Italy). The AE extract was obtained by mixing artichoke leaves with water (ratio 1:10) for 1 h, after which it was filtered and evaporated. The content of total dicafeylquinic acids in artichoke extracts, expressed as chlorogenic acid, was determined according to the Official Pharmacopoeia of the Italian republic (FUI, XII 2008). In detail, 0.2 g of the extract was dissolved in 5 mL of distilled water. After adding 10 mL of a lead acetate solution (95 g/L), the sample was homogenized and centrifugated for 15 min at 2000× *g*. The supernatant was discarded, while the residue was dissolved in a 2.5 mL acetic acid solution containing 12% of glacial acetic acid and 12.5 mL sulfuric acid solution 1N containing 5.6% sulfuric acid, shacked for 1 h, and then diluted in 50 mL of distilled water. After centrifugation (15 min at 2000× *g*), 2 mL of the supernatant was diluted in 50 mL of methanol, and the absorbance was spectrophotometrically measured at 325 nm while using a methanol blank sample. The content of the total dicafeylquinic acids expressed as chlorogenic acid was calculated with the following equation:A×2500/485×m×2
where *A* represents the value of absorbance, and *m* is the mass (in grams) of artichoke extract in the solution.

### 2.3. Bread Preparation

Gluten-free bread (DY 200) batches were manufactured at the pilot plant of the Department of Soil, Plant, and Food Science of the University of Bari (Italy). In detail, four batches were used: (i) baker’s yeast gluten-free bread produced without the addition of sourdough and artichoke extract (YB)’ (ii) baker’s yeast gluten-free bread produced with the addition of artichoke extract (YB-AE)’ (iii) tII-SD gluten-free bread (SB)’ and (iv) tII-SD gluten-free bread with the artichoke extract (SB-AE).

The sourdough bread (SB and SB-AE) batches were obtained according to a double-fermentation process accounting for, first, the production of tII-SD (fermentation for 24 h at 30 °C), which was then added (30% *w*/*w* of dough) in a dough containing rice flour, water, baker’s yeast (1.25% *w*/*w*). In AE-containing bread batches (YB-AE and SB-AE), the powder extract of artichoke leaves was also added (6% *w*/*w*) as a substitute for the rice flour.

All the doughs were subjected to an incubation step (1.5 h at 30 °C) before being baked at 210 °C for 30 min (Combo 3, Zucchelli, Verona, Italy).

### 2.4. Bread Characterization

#### 2.4.1. Predicted Glycemic Index (pGI)

The in vitro gastrointestinal digestion of bread was performed following previous procedures [39]. Aliquots (5 g) were subjected to the enzymatic process (pancreatic amylase and pepsin–HCl) to measure the released glucose content using a D-fructose/D-glucose Assay Kit (Megazyme; Wicklow, Ireland). In addition, digests were dialyzed (membrane cut-off: 12,400 Da) for 180 min. Dialysate aliquots were collected every 30 min, and further treated with amyloglucosidase, to determine the free glucose using the above-mentioned kit. A control white wheat bread sample was used to estimate the hydrolysis index (HI = 100). As stated by Goñi et al. (1997) [40], the pGI was calculated with the following equation:pGI=0.549×HI+39.71

Each sample was analyzed in triplicate.

#### 2.4.2. Antioxidant Activity

The bread’s antioxidant activity was measured based on DPPH and ABTS assays. As previously described [38], methanolic extracts (MEs) were obtained from each bread sample. Aliquots (5 g) of each sample were mixed with 50 mL of 80% methanol to obtain MEs. Under stirring conditions, the mixture was purged with nitrogen stream for 30 min and then centrifuged (4600× *g* for 20 min). The MEs were transferred into test tubes and stored at 4 °C until further processing. The DPPH (2,2-diphenyl-1-picrylidrazyl) assay was performed in cuvettes using a spectrophotometer and adding 50 µL of each ME and 950 µL of the DPPH (0.08 mM) ethanolic solution. After 30 min in the dark, the decrease in absorbance was measured at 517 nm using an Agilent Cary 60 spectrophotometer (Cernusco, Italy).

Then, 25 mL of radical ABTS (2,2’-azino-bis (3-ethylbenzothiazoline-6-sulfonic acid)) solution (7 mM) was spiked in 440 μL of K_2_S_2_O_8_ (140 mM) and kept in the dark at room temperature for 16 h. The working solution, diluted with sterile water, was prepared to obtain a final absorbance at 734 nm equal to 0.8 ± 0.02 [41]. Each ME (50 μL) was added to 950 μL of ABTS reagent and the decrease in absorbance was measured (734 nm) after incubation (8 min).

For both assays, the results are expressed as μmol of Trolox equivalent (TE)/g of extract. Each sample was analyzed in triplicate.

### 2.5. In Vitro Bread Digestion and Simulated Colonic Fermentation

For the simulated in vitro digestion of bread samples, we assessed the enzymatic contribution of fluids participating in oral, gastric, and intestinal phases by following standardized procedures [42].

To constitute the fecal medium, four fecal samples were equally mixed and added at 20% (*w*/*w*) in a sterile NaCl solution (0.9% *w*/*v*). Using filtering bags, these were homogenized with a stomacher (Bag Mixer, Interscience International; Roubaix, France) for 3 min, and after recovering the residual over the filter, the suspension was centrifuged (10,000× *g*, 10 min, 4 °C). The pellet was discarded, while the supernatant was supplemented (K_2_HPO_4_•2 g/L, C_2_H_3_NaO_2_•5 g/L, C_6_H_17_N_3_O_7_•2 g/L, MgSO_4_•0.2 g/L, MnSO_4_•0.05 g/L, glucose•2 g/L, inulin•4 g/L, fructo-oligosaccharides•4 g/L, Tween 80 polysorbate•1 mL/L) as previously detailed [21] and then sterilized (121 °C, 20 min) after the pH adjusting to 7.0 with a NaOH (6 M) solution. Except for inulin and fructo-oligosaccharides, which were purchased from Farmalabor S.r.l., all other mentioned reagents were purchased from Sigma-Aldrich (St. Louis, MO, USA).

By combining different concentrations of the fecal medium with each digested bread sample, four different fecal media were obtained and used as substrates for the fecal slurry fermentation.

Within 1 h of delivery, fresh fecal samples of one healthy volunteer were added at 32% (*w*/*v*) in distilled water, and this fecal slurry was then added at 20% (*v*/*v*) to the four different fecal media to constitute fecal batches [43].

The fecal batches were anaerobically incubated at 37 °C under gently stirring conditions (100× *g*) for 42 h, while an intermediate aliquot (5 g) was collected from each fecal batch at 20 h of incubation.

### 2.6. Microbiota and Volatilome Profiling of Fermented Fecal Batches

#### 2.6.1. Enumeration of Microbes from Fecal Batches

An aliquot (5 g) of each fermented fecal batch was mixed with 45 mL of the sterilized NaCl solution (0.9% *w*/*v*) and homogenized. Decimal dilutions were carried out. Viable bacterial cells were counted in the following culture media: plate count agar (total aerobes); Wilkins–Chalgren anaerobe agar (total anaerobes); MRS agar (LAB); M17 agar (streptococci); violet red bile glucose agar (coliforms); reinforced *Clostridium* agar (clostridial microbes); and modified *Bifidobacterium* agar (Becton Dickinson; Le Pont de Claix, SA, France) (fecal bifidobacteria). Except for modified *Bifidobacterium* agar, other media were purchased from Oxoid Ltd. (Basingstoke, Hampshire, England). Except for Wilkins–Chalgren anaerobe agar and modified *Bifidobacterium* agar, other media were aerobically incubated. For culturing conditions, the adopted times and temperatures varied according to manufacturers’ instructions.

#### 2.6.2. Fecal Volatile Organic Compound Profiling of Fecal Batches

An aliquot corresponding to 1 g of each fermented fecal batch was used to characterize the volatile organic compound (VOC) profiles. The samples were equilibrated for 10 min at 60 °C. The SPME fiber (divinylbenzene/Carboxen/polydimethylsiloxane) was exposed to each sample for 40 min. The volatile organic compounds (VOCs) were thermally desorbed by immediately transferring the fiber into the heated injection port (220 °C) using a Clarus 680 (Perkin Elmer, Beaconsfield, UK) gas chromatograph equipped with an Rtx-WAX column (30 m × 0.25 mm i.d., 0.25 µm film thickness) (Restek) and coupled to a Clarus SQ8MS (PerkinElmer) [44]. Then, 10 µL of 4-methyl-2-pentanol (final concentration of 9.9 mg/L) was added as an internal standard. Each generated chromatogram was analyzed for peak identification using the National Institute of Standard and Technology 2008 (NIST) library. A peak area threshold of >1,000,000 and 85% or greater probability of match was used for VOC identification, followed by the manual visual inspection of the fragment patterns when required. In addition, 4-methyl-2pentanol (internal standard) was used to quantify the identified compounds via the interpolation of the relative areas versus the internal standard area.

### 2.7. In Vitro Assays of Digested Bread Samples on Cell Cultures

#### 2.7.1. Cell Culture Conditions

To perform the experiments, two different human cell lines were used, and both were supplied by the National Institute for Cancer Research of Genoa (Italy). Specifically, the cell lines were the Caco-2 (colon adenocarcinoma) ICLC HTL97023 and the human keratinocyte NCTC 2544.

The Caco-2 cell line was cultured in a DMEM GLUTAMAX medium (Gibco, Thermo-Fisher Scientific, Inc.; Waltham, MA, USA) supplemented with 10% fetal bovine serum (FBS, Lonza Bioscience; Walkersville, MD, USA), 2 mM L-glutamine (Gibco, Thermo-Fisher), a 1% HEPES solution (Sigma-Aldrich), 1% of penicillin (10,000 U/mL)/streptomycin (10,000 μg/mL), and 0.1% gentamicin mixture (Lonza Bioscience and Sigma-Aldrich, respectively), maintained in 25 cm^2^ culture flasks at 37 °C in 5% CO_2_. Every two days, after washing with phosphate-buffered saline (PBS) free of Ca^2+^ and Mg^2+^ (Lonza Bioscience)**,** confluent cultures were split 1:3–1:6 using Trypsin/EDTA (Lonza Bioscience) and seeded at 2–5·10^4^ cells/cm^2^, 37 °C, 5% CO_2_.

The human keratinocyte cell line was cultivated in an RPMI medium (Sigma-Aldrich) supplemented with 10% FBS (Lonza Bioscience), 2 mM L-glutamine (Gibco, Thermo-Fisher), a 1% penicillin (10,000 U/mL)/streptomycin (10,000 μg/mL) mixture (Lonza Bioscience) and maintained in 25 cm^2^ culture flasks at 37 °C in 5% CO_2_. Every two days, after washing with PBS (Lonza Bioscience), confluent cultures were split 1:3–1:6 using trypsin/EDTA (Lonza Bioscience).

#### 2.7.2. MTT Assay

Before proceeding with the assay, Caco-2 cells were placed into 96-well plates and incubated with the DMEM GLUTAMAX medium at 37 °C under 5% CO_2_. After 24 h of incubation, supernatants from fecal batches were added at different concentrations (0.001, 0.01, and 0.1 mg/mL) in the above-detailed culture medium and incubated at 37 °C, 5% CO_2_, for 16 h. The MTT (3-(4,5-dimethylthiazol-2-yl)-2,5-diphenyltetrazolium bromide) assay was conducted as previously detailed [45], using an MTT solution (Sigma-Aldrich) and a scanning multi-well spectrophotometer (mod. ELX808, BioTech, Agilent Technologies; Santa Clara, CA, USA), which was set at a wavelength of 570 nm with a reference filter of 630 nm.

The experiments were carried out in triplicates. The Caco-2 cell’s viability was compared against negative controls, which were Caco-2 cells incubated with the culture medium without supernatants, and it was expressed as percentages.

#### 2.7.3. Protection from Induced Oxidative Stress

The viability of H_2_O_2_-stressed human keratinocyte cells was also determined with the MTT assay as reported elsewhere [46]. The experimental setting considered the contribution given by fecal batch supernatants added at different concentrations (0.1, 0.01, and 0.001 mg/mL) to the culture medium and incubated at 37 °C, 5% CO_2_ for 16 h. After the removal of the culture medium, cells were exposed to 100 µL/well of 400μM hydrogen peroxide at 37 °C, 5% CO_2_ for 2 h. The plate was carefully wrapped with foil allowing air circulation while covering the whole plate to minimize light exposure. Following hydrogen peroxide treatment, the medium was removed and replaced by 100 µL/well of the MTT salt solution and incubated at 37 °C, 5% CO_2_ for 3 h.

A positive control sample (culture medium without supernatants) and a negative control sample (cells not exposed to H_2_O_2_) were also used.

#### 2.7.4. Cytokines Assay

The anti-inflammatory contribution of supernatants from digested bread samples was assessed, estimating the levels of tumor necrosis factor-alpha (TNF-α) and interleukin 1-β expression in Caco-2 cells. Cells were exposed to 10 µg/mL of lipopolysaccharides from *E. coli* (LPS, Sigma-Aldrich). LPS exposure was assessed with supernatants (experimental theses) or without supernatants (positive control) added at a final concentration of 0.01 μg/mL in the culture medium. After 24 h of incubation at 37 °C and 5% CO_2_, RNA for qPCR analysis was extracted. A negative control (cells + culture medium) was also processed. The cDNA was synthesized from 2 μg of mRNA template in a 20 μL reaction volume using the PrimeScript RT-PCR Kit (Takara, Japan). Tumor necrosis factor-alpha (TNF-α Hs00174128_m1), interleukin-1-beta (IL-1β Hs01555410_m1), and human glyceraldehyde-3-phosphate dehydrogenase (GAPDH, used as housekeeping gene; Hs999999_m1) primers were purchased from Applied Biosystems (Thermo-Fisher Scientific, Inc.). The amplification and detection of qPCR amplicons were carried out using a Stratagene Mx3000P^TM^ Real-Time PCR System (Agilent Technologies Italia S.p.A.; Milan, Italy). The samples were processed in duplicates.

### 2.8. Statistics

As appropriate, data were summarized using descriptive statistics, that is, means ± standard deviations (±SD), means with 95th percentile of the confidence interval (95%CI), or relative frequencies (percentages). The one-way analysis of variance (ANOVA) assessed significant differences (*p*-value < 0.05) between the mean values. A normalized matrix by means and standard deviation scaling (Z-score) was used for the principal component analysis (PCA) computed in the R-environment using the “FactoMineR” package (Multivariate Exploratory Data Analysis and Data Mining) version 2.4 available in the CRAN repository. The same package was also used for the heatmap and clustering; the latter was based on Euclidean distance and Ward’s similarity.

## 3. Results

### 3.1. Sourdough Characterization

Acting as a bread ingredient, the selected tII-SD was profiled in detail. After 24 h of incubation, the selected tII-SD showed a LAB cell density of 8.6 ± 0.2 log (UFC/g), the pH was 4.14 ± 0.01, while when determining the TTA, 4 ± 0.05 mL of 0.1 NaOH solution was required to reach the pH value of 8.3. Concerning the fermentation quotient, lactate and acetate approximately showed a ratio of 3:1, which were assessed with concentrations of 7.93 ± 0.2 and 2.8 ± 0.1 g/100 g of tII-SD, respectively.

### 3.2. Bread Characterization

#### 3.2.1. pGI

The predicted glycemic index (*pGI*) was calculated in digested samples. The value of *pGI* ranged from 62.8 to 111.3 (Figure 1A) and these were respectively assessed in SB and YB-AE digested bread samples. Clear tendencies toward decreased *pGI* were observed in bread batches with tII-SD. In fact, both SB and SB-AE showed lower (*p* ≤ 0.004) *pGI* values than those found in YB and YB-AE (Figure 1B). Although no difference existed between YB and YB-AE, the presence of AE significantly increased the *pGI* in SB-AE.

#### 3.2.2. Antioxidant Activity

The free radical scavenging activity was assessed in bread samples to investigate antioxidant properties (Figure 2). The pairwise comparison between bread samples with or without AEs showed that DPPH values were more than 15-fold increased (*p* < 0.001) in AE-supplemented bread samples, i.e., SB-AE and YB-AE. The comparison between SB-AE and YB-AE showed that the former had the highest value of DPPH (*p* = 0.02).

The free radical scavenging was also evaluated for the radical cation ABTS, confirming the AE contribution in improving the antioxidant activity by about 10-fold (*p* < 0.001) (Figure 2). Additionally, the SB-AE combination led to a higher (*p* = 0.002) antioxidant activity than YB-AE.

### 3.3. In Vitro Effects of Digested Bread

#### 3.3.1. Culturable Microbiota of Fecal Batches

Simulating the colonic fermentation, the effectiveness of the digested bread samples in affecting the viability of fecal microbiota cells was tested in batches fermented for 20 and 42 h. Based on principal component analysis (PCA), no clear tendencies emerged (Figure 3). Based on the first principal component (PC1), the samples clustered according to the fermentation time because of T20 samples showed positive PC1 values, while T42 samples showed negative PC1 values. Additionally, the variance explained by additional PCs was determined (Appendix A). In the PCA plot (Figure 3), the baseline samples (T0) showed the lowest cell viability. Even the viability of *Enterobacteriaceae* was remarkably affected at T0. At T20, the samples showed increased viability of total bacteria (TBC), *Enterobacteriaceae,* and LAB. In comparison, the viability of lactic streptococci, clostridial, and bifidobacterial cells was increased at T42. Few differences were observed with AE supplementation. At T42, *Bifidobacterium* and clostridial microbes were mainly harbored by YB, and the addition of AE did not affect this scenario. In parallel, SB mainly harbored LAB (including streptococci), without any significant differences with the addition of AE.

#### 3.3.2. Fecal Volatile Organic Compounds in Fecal Batches

The metabolic profile characterization of fecal batches after 20 h and 42 h of incubation was based on qualitative and quantitative differences in VOCs. Overall, 59 volatile metabolites were identified and classified into the following chemical classes: alcohols (5), aldehydes (10), esters (6), hydrocarbons (6), indoles (2), ketones (8), organic acids (14), phenols (2), and terpenes (2). Four additional compounds not belonging to the listed classes were identified, i.e., 3-ethyl-3-methylheptane; 1H-pyrrole-2,5-dione; 3-ethyl-4-methyl,8-methylnonanoic acid; and γ-dodecalactone.

The unsupervised PCA analysis showed 64% variance (PC1 44.5% and PC2 19.4%) clearly differentiated the fecal batches based on PC1, which differentially allocated control and experimental theses on the PCA score plot, and PC2, which mainly supported differences occurring between the two sampling times (T20 and T42) (Appendix A). To better inspect the VOC differences among the experimental theses, an intergroup comparison for each sampling time was run. Eleven and thirty-four compounds significantly differed at T20 and T42, respectively (Appendix A, respectively). The statistically different compounds were used to compute a hierarchical clustering analysis, which revealed how the fecal batches clustered after 20 h of incubation based on the use of tII-SD (Appendix A). Differently, after 42 h of incubation, the fecal batches were grouped in two different clusters based on the AE addition (Figure 4). In detail, cluster 1, including alcohols (i.e., ethanol; phenylethyl alcohol; and 1-propanol) organic acids (i.e., propanoic acid; 2-methyl propanoic acid; butanoic acid; pentanoic acid; hexanoic acid; and heptanoic acid), phenols (i.e., phenol and p-cresol), benzaldehyde and 1H-indole,3-methyl-(skatole), showed higher Z-scores in AE-containing bread samples than those not containing AE. Within cluster 2, subcluster 2a included one alcohol (1-butanol) and three aldehydes (tetradecanal; octadecanal; and dodecanal) with the highest and the lowest Z-score in SB and SB-AE, respectively. In comparison, subcluster 2b included VOCs with the highest Z-scores in SB-AE, mainly cyclohexanecarboxylic acid; hydrocinnamic acid; hexanoic acid ethyl ester; 1H-Pyrrole-(2, 5-dione, 3-ethyl-4-methyl); and 8-methylnonanoic acid.

### 3.4. Ex vivo Effects of Digested Bread Samples after Simulated Colonic Fermentation

#### 3.4.1. Cytotoxicity

By using Caco-2 cells, the cytotoxicity of supernatants collected from fecal batches was evaluated setting as 100% of cell viability the value provided by Caco-2 cells incubated with the culture medium without supernatants (CNT; Appendix A). The T0 fecal batch (fecal microbiota inoculum) displayed the highest value of cell viability, while the supernatants from the fermented batches reduced cell viability between 82.7% and 99.2%, with the lowest observed in YB-AE-T20 and the highest in SB-AE-T20 (Appendix A). However, no statistically significant differences (*p* > 0.05) existed comparing supernatant-containing theses and CNT.

The same assay was then applied to H_2_O_2_-stressed human keratinocytes to assess the presumptive protecting role of AE against the stressor agent. Again, the 100% human keratinocyte viability corresponded with the values provided by cells incubated with the related culture medium. Thus, the exposure to H_2_O_2_ significantly (*p* < 0.001) decreased cell viability to 68.34%, whereas on average, the supernatant addition to the culture medium led to increased cell viability despite the concomitant exposure to H_2_O_2_ (Figure 5A). The differences were investigated by comparing values of keratinocytes exposed to H_2_O_2_ (without supernatant addition) and theses containing different concentrations of supernatant in the culture medium (0.001, 0.01, and 0.1 mg/mL; Figure 5B–D, respectively). Evaluating the supernatants collected after 20 h of fecal batch fermentation, except for YB-T20 and SB-T20 added at 0.1 mg/mL (Figure 5D), the others significantly increased the human keratinocyte viability. Differently, using the T42 supernatants, only those derived from the fecal batch fermentation of AE-containing bread samples demonstrated the capability to protect cells from H_2_O_2_ stressor agents.

#### 3.4.2. Anti-Inflammatory Effects

Based on TNF-α and IL1-β expressions, Caco-2 cells were also used to estimate the anti-inflammatory activity of the supernatants from colonic fermented bread samples, while the LPS from *E. coli* was used as a proinflammatory trigger (positive control). Compared with the positive control, a significant decrease (*p* < 0.05) in TNF-α was detected in SB-T42, SB-AE-T20, and SB-AE-T42 (Figure 6A). Interestingly, SB-AE-T42 did not differ (*p* > 0.05) from the negative control, i.e., the cells not exposed to LPS.

Compared with the positive control, both supernatants of SB-AE (T20 and T42) significantly decreased (*p* < 0.05) the IL1-β expression, and both did not significantly differ (*p* > 0.05) from the negative control (Figure 6B).

## 4. Discussion

Following the definition of “functional food” [47], “dietary adjuvant therapies” rely on personalized treatments with functional food added to diets to achieve desired outcomes. The purpose of this study was to modify a traditional food product consumed daily worldwide to support patients with GrDs and those with proven CeD. We recently revised the literature in the field of CeD-related metabolomics, also ascertaining the differences affected by daily gluten intake or deprivation [9]. The mentioned study revealed that markers of oxidative stress were frequently altered in CeD patients at diagnosis, and this scenario did not return to healthy ranges despite adhering to GFD [16]. Given the need for antioxidant supplementation, various vegetables can be used as natural sources of phenols, and in this regard, artichoke is an optimal candidate [17,48]. Due to the high rates of artichoke waste from industrial processes [49], recent studies focused on the development of innovative methods allowing for phenol recovery and providing promising outcomes. Jiménez-Moreno and co-workers recovered around 2500 ppm of phenolic components from the waste material of artichoke bracts, outer leaves, and stems [25]. By setting a water-based extraction protocol, Garcia-Castello et al. [26] revealed that it was possible to recover 60% of the polyphenolic quantity and 56% of antioxidant activity from the solid waste of artichoke by-products. Using ultrasound-assisted procedures, Turker et al. [27] extracted a remarkable quantity of polyphenols from artichoke waste in a shorter time than traditional extraction methods. Recent advances in recovering phenolic compounds from artichoke by-products also allowed researchers to obtain phenol-enriched food products, e.g., fresh pasta [50,51]. In these studies, the artichoke extract’s phenolic profile was characterized as a large spectrum of different compounds, whereas we limited our study to the assessment of dicafeylquinic acid content expressed as a chlorogenic acid percentage. Differently, the focus of our work was the evaluation of the potential functional activities that bread can directly provide to the host’s cells or via gut microbiota fermentation. To achieve this goal, we tested fermented bread batches with fecal microbiota on two different cell lines because, while Caco-2 cells have been widely used to perform cytokine assays [52], it was suggested that this cell line was not the optimal choice to test the oxidative stress damage [53]. Moreover, based on both dietary adjuvant therapy and functional food concepts, we here combined tII-SD with AE to obtain a healthier baked product suitable for GrD patients. Since microbial fermentations represented a way to metabolize sugar-containing substrates, sourdough-based bread products are well known for their reduced GI [33]. LAB metabolism also represents a tool to hydrolyze wheat and rye proteins responsible for cereal allergy [54]. Thus, the sourdough technique is additionally advantageous due to the healthier features offered by its derived products [37]. Furthermore, bacterial fermentation can also be exploited to obtain clean-label bread with ameliorated pretechnological properties [55].

Herein, the AE increased the antioxidant activity of the related bread products more than 10-fold. This compound can exert a broad range of beneficial effects targeting different organs and systems, including the nervous, cardiovascular, gastrointestinal, renal, and hepatic systems [56]. Interestingly, the combination of AE with tII-SD further increased the resulting scavenging activity compared with that exerted by YB-AE. In sourdough bread samples, LAB metabolized chlorogenic acid, increasing its availability, as previously found after lactic acid fermentation [57,58]. Although the chemical class of phenols also encompasses antimicrobial compounds [59,60], compared with other probiotics (e.g., bifidobacteria), a higher concentration of chlorogenic acid is needed for inhibiting LAB [61] even though this should be considered under a strain-dependent manner [62]. Thanks to its sequestering behavior, as well as oxygen- and metal-reducing capacity [63], the AE protected human keratinocytes against oxidative stress, and although we observed significantly different scavenging activity, both YB-AE and SB-AE contained enough AE to improve the viability percentages of human keratinocytes.

Differently, the highest anti-inflammatory effectiveness was found only using supernatants from SB-AE samples because only these samples displayed a significant reduction in both TNF-α and IL1-β expressions under all the tested conditions in Caco-2 cells exposed to LPS. The direct anti-inflammatory properties of chlorogenic acid are well recognized [56,63,64,65], while the present study provides evidence of how AE fermentation with sourdough biotechnology allowed for the elimination of differences with the negative control, i.e., the cells not exposed to LPS.

Herein, VOCs were profiled to extensively evaluate the presence of specific metabolites with healthy properties in the residual from fecal microbiota fermentation. Under simulated conditions, 20 h of fecal fermentation distinguished SBs from YBs with the presence of aldehydes and hydrocarbons in the latter group without significant differences resulting from the addition of AE. High scores of hydrocinnamic acid, instead, featured only SB-AE-T20. This compound is one of the most representative phenolic compounds of artichoke and, consequently, is present in its waste despite the different factors (plant variety, climate, and harvest time) that can contribute to modifying the overall plant chemical composition [66,67]. In a recent study, hydrocinnamic acid was also the metabolite more representative of control volunteers than patients with the atherosclerotic disease [68]. Moreover, the same study also demonstrated a positive correlation between hydrocinnamic acid and a bacterial pattern accounting for various health-promoting saccharolytic microbes [68]. The VOC-based scenario partially changed following an additional 22 h of incubation. In fact, at 42 h, a wide spectrum of organic acids, phenols, and indoles differently featured the AE-containing samples. Typically, indoles and phenols are derived from the metabolism of aromatic amino acids such as tryptophan, tyrosine, and phenylalanine by the gut microbiota [69,70]. In turn, tryptophan-derived metabolites have been reported to have a pivotal role in the gut microbiota–host crosstalk, acting as aryl hydrocarbon receptor (AhR) ligands and agonists and, therefore, becoming responsible for a cascade of immunomodulatory events (e.g., the modulation of intraepithelial lymphocytes, T cells, and group-3 innate lymphoid cells) [71,72,73]. The resulting scenario led to the expression of interleukin-17 (IL-17) and IL-22 that can be intended as beneficial outcomes for GrD-suffering patients [74]. At T42, SB-AE once again was found to have the highest hydrocinnamic acid Z-score, and it also displayed the highest score of cyclohexanecarboxylic acid. This carboxylic acid, contained in several medicinal plant extracts, has been exploited for multiple biological applications (e.g., antioxidant, antimicrobial, antidiabetic, cardioprotective, and anticancer activity) [75]. However, although AE was also added to YB, neither hydrocinnamic acid nor cyclohexanecarboxylic acid was reported to have similar results in YB-AE, suggesting the occurrence of a positive interaction between tII-SD microbiota and AE-metabolism. This can explain the difference found in terms of anti-inflammatory activity in Caco-2 cells exposed to LPS.

## 5. Conclusions

These results support the possible application of artichoke leaf extract as a functional ingredient for developing novel gluten-free products with improved biological properties. Sourdough biotechnology, pivotal for reducing the glycemic index in leavened goods, allows for avoiding the occurrence of those problems usually resulting from gluten-free bakery products. The artichoke leaf extract, instead, allows for providing a phenolic supplementation for patients with celiac disease, thus reducing the level of cellular oxidative stress usually found in these patients. Based on in vitro and ex vivo experiments, we demonstrated a significant amelioration of both oxidative stress and proinflammatory cytokine expression in cells.

Therefore, this research further supports the application of novel and personalized therapies based on nutritional management, and these data can be integrated into the field of research exploring the recombination of various dietary components to design functional and clean-label products.

## Figures and Tables

**Figure 1 antioxidants-12-00845-f001:**
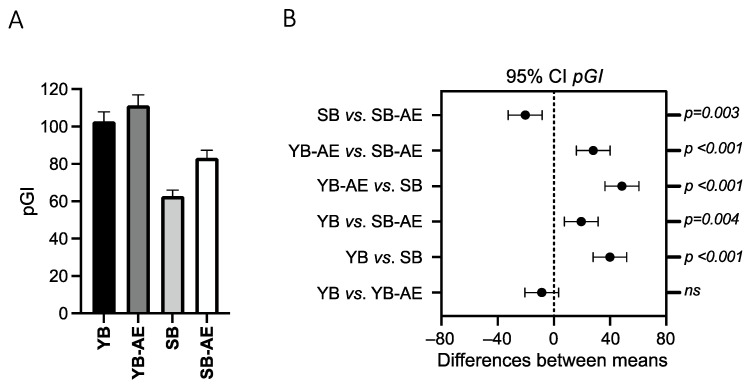
The predicted glycemic index (*pGI*) stacked bars (**A**) of digested bread samples prepared with or without type-II sourdough (S and Y, respectively) and with or without a powdered artichoke extract (AE). Statistical comparison (**B**) shows the 95% confidence interval (CI) and related *p*-values (*p*). *ns*: not significant.

**Figure 2 antioxidants-12-00845-f002:**
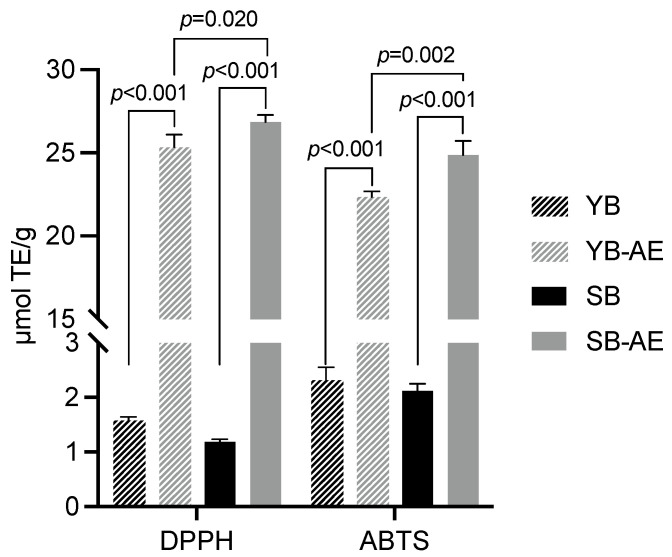
Radical scavenging activity, based on DPPH and ABTS assays, of bread samples. Results are expressed as μmol of Trolox equivalent (TE)/g. Bread samples were prepared with or without type-II sourdough (Y and S, respectively) and with or without a powdered artichoke extract (AE). Error bars are shown. Significant differences (*p*-value < 0.05) are shown as exact *p*-values.

**Figure 3 antioxidants-12-00845-f003:**
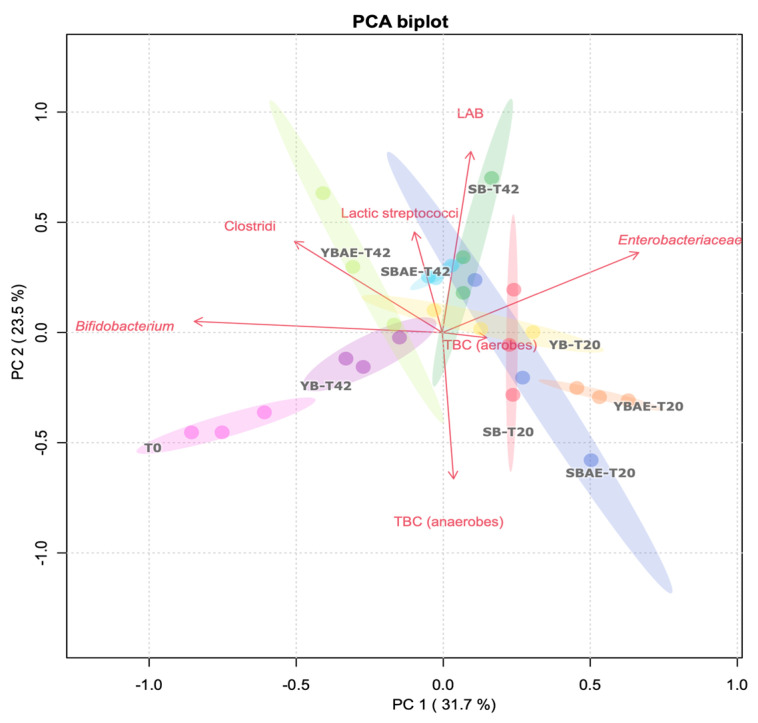
Biplot of the principal component analysis (PCA) of viable microbial cell densities in fermented fecal batches. Both score and loading plots are shown. The score plot shows fermented fecal batches different for the used bread digest (yeast bread, YB; yeast bread with artichoke extract, YB-AE; sourdough bread, SB; and sourdough bread with artichoke extract, SB-AE) and for the time of fermentation (20 and 42 h, T20 and T42, respectively). Based on normalized values, the loading plot shows the cell densities of the plated microbial groups. Clouds (weighted on the 95th percentile of the confidence interval of three replicates) have different colors for both digested bread and the time of fermentation. Cloud colors represent baseline (that is, the inoculum labeled as T0, pink), YB-T20 (yellow), YB-T42 (purple), YB-AE-T20 (orange), YB-AE-T42 (light green), SB-T20 (red), SB-T42 (green), SB-AE-T20 (blue), and SB-AE-T42 (light blue). Abbreviations: total bacterial count (TBC); lactic acid bacteria (LAB).

**Figure 4 antioxidants-12-00845-f004:**
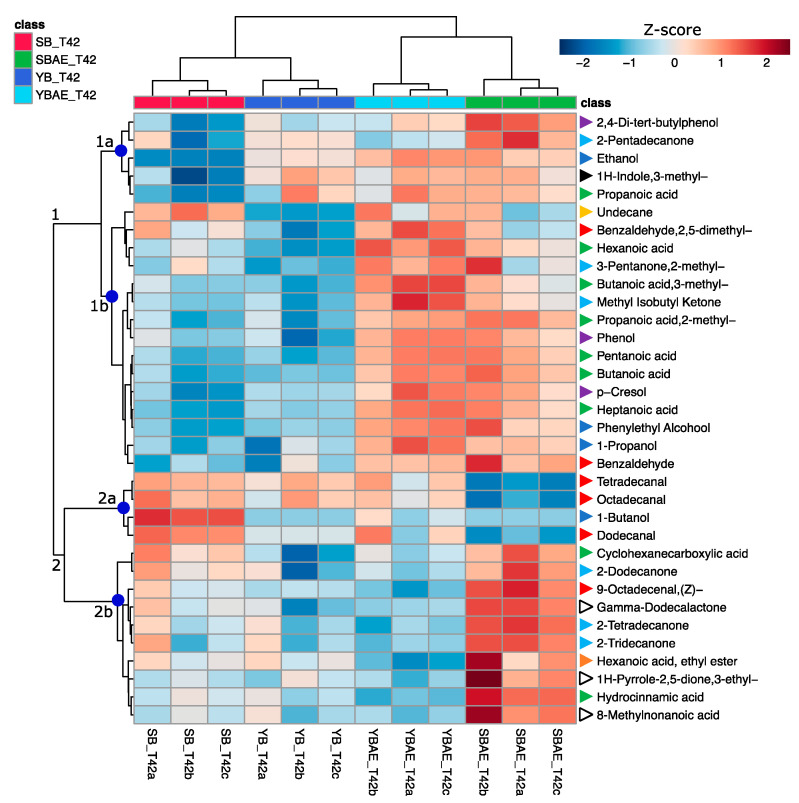
Heatmap with clustering (Ward’s method) of significant (*p*-value < 0.05; one-way ANOVA) volatile organic compounds (VOCs) found after 42 h of fermentation in fecal batches of digested bread samples (yeast bread, YB; yeast bread with artichoke extract, YB-AE; sourdough bread, SB; and sourdough bread with artichoke extract, SB-AE). Differently colored triangles indicate the chemical class: phenols (purple), ketones (light blue), alcohols (dark blue), indoles (black), organic acids (green), hydrocarbons (yellow), aldehydes (red), esters (orange), and others (white).

**Figure 5 antioxidants-12-00845-f005:**
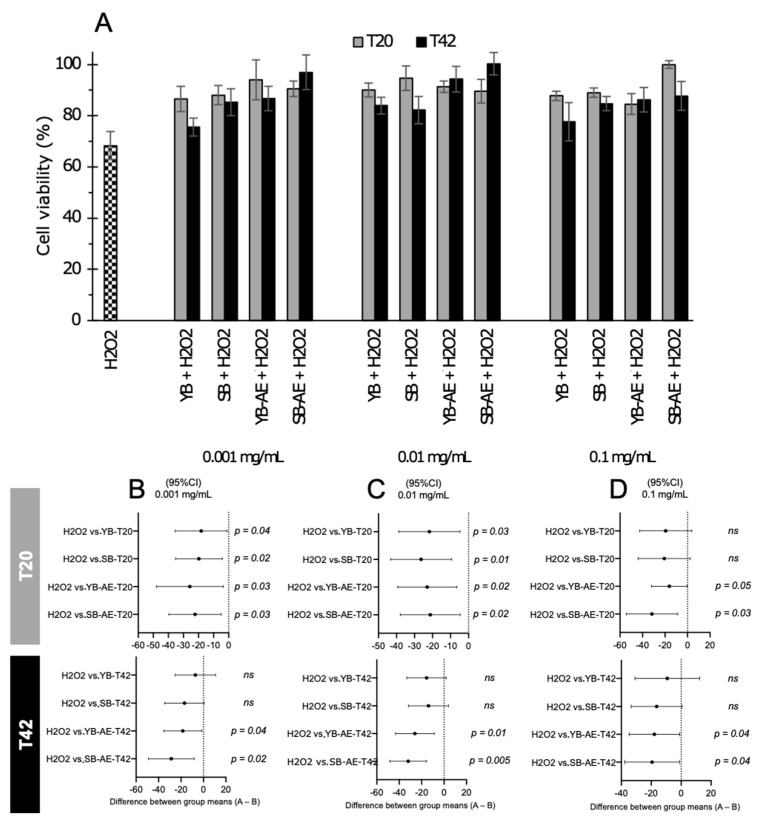
Protective effect of digested bread on human keratinocyte cell viability against oxidative stress induced by hydroxide peroxide (H_2_O_2_). Bread batches were prepared with or without type-II sourdough (YB and SB, respectively) and with or without a powdered artichoke leaf extract (AE). Supernatants from fecal batches derived from fermented bread digestion were recovered at different times (20 and 42 h, T20 and T42, respectively) and tested at different concentrations (0.1–0.001 mg/mL). The 100% of cell viability corresponded with values of cells cultured in the medium under optimal conditions. The viability of cells exposed to the H_2_O_2_ stressor agent is reported (**A**). Data are mean values of three independent experiments analyzed twice. Error bars are shown in the bar plot. The protective role accounted for a significant difference (*p*-values < 0.05, one-way ANOVA test) between cells exposed to the H_2_O_2_ stressor agent and cells incubated with different concentrations of supernatants in the culture medium (0.001, 0.01, and 0.1 mg/mL: (**B**–**D**), respectively). *ns*, not significant.

**Figure 6 antioxidants-12-00845-f006:**
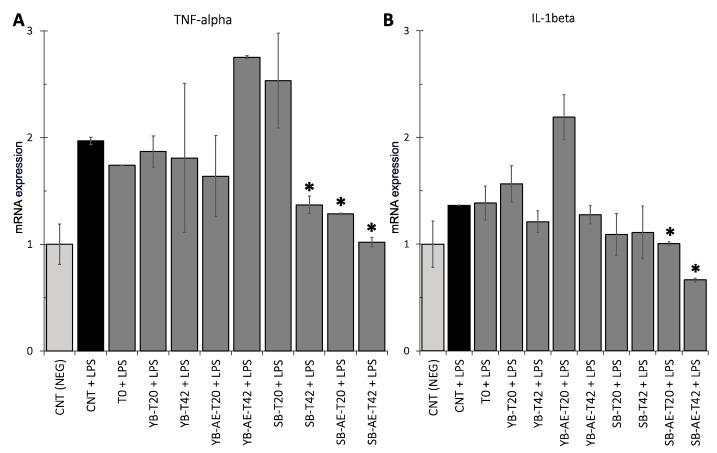
Tumor necrosis factor-alpha (TNF-α) (**A**) and interleukin 1-beta (IL1-β) (**B**) expression in Caco-2 cells exposed to LPS from *E. coli* trigger. Analyses were carried out in two independent experiments. The negative control (NEG CNT) was the medium for Caco-2 cell culture, while the positive control was CNT + LPS. Error bars are shown. Significance, labeled with “*”, was reached with a *p*-value < 0.05 comparing samples against CNT + LPS (one-way ANOVA test).

## Data Availability

Not applicable.

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
