# Peer review of "Gluten-Free Bread Enriched with Artichoke Leaf Extract In Vitro Exerted Antioxidant and Anti-Inflammatory Properties"

_antioxidants, 2023, doi:10.3390/antiox12040845_

Round 1

Reviewer 1 Report

The manuscript "Gluten-free breads enriched with artichoke leaf extract in vitro exerted antioxidant and anti-inflammatory properties" deals with an interesting way to enrich the diets of celiacs with a functional product. The study is well done but an essential element is missing in the description of the materials and therefore in the discussion of the results is lacking. In fact, the authors do not describe how they obtained the artichoke extract and above all they have not given a detailed chemical description. In this regard, the authors are invited to add the characterization of the extract used and to consider two publications for a comparison in this regard:

Pasqualone, A.; Punzi, R.; Trani, A.; Summo, C.; Paradiso, V.M.; Caponio, F.; Gambacorta, G. Enrichment of fresh pasta with antioxidant extracts obtained from artichoke canning by-products by ultrasound-assisted technology and quality characterisation of the end product. Int. J. Food Sci. Technol. 2017, 52, 2078–2087.

Amoriello, T.; Mellara, F.; Ruggeri, S.; Ciorba, R.; Ceccarelli, D.; Ciccoritti, R. Artichoke By-Products Valorization for Phenols-Enriched Fresh Egg Pasta: A Sustainable Food Design Project. Sustainability 2022, 14, 14778. https://doi.org/10.3390/su142214778

Further changes must be entered.

Equalize the number of significant figures in the means and SDs and in the p-values in 3.1 and 3.1.2, figures 1b and 2.

Figure 3 and paragraph 3.2.1: Depicting only two PCs in the PCA when the explained variance is 55% is incorrect. It is suggested to the authors to use at least 3 components, or, better, to tabulate the contributions of the different variables to the loadings. Therefore, the description of the results in paragraph 3.2.1 needs to be rewritten.

Author Response

#R1.

The manuscript "Gluten-free breads enriched with artichoke leaf extract in vitro exerted antioxidant and anti-inflammatory properties" deals with an interesting way to enrich the diets of celiacs with a functional product. The study is well done but an essential element is missing in the description of the materials and therefore in the discussion of the results is lacking.

In fact, the authors do not describe how they obtained the artichoke extract and above all they have not given a detailed chemical description. In this regard, the authors are invited to add the characterization of the extract used and to consider two publications for a comparison in this regard:

Pasqualone, A.; Punzi, R.; Trani, A.; Summo, C.; Paradiso, V.M.; Caponio, F.; Gambacorta, G. Enrichment of fresh pasta with antioxidant extracts obtained from artichoke canning by-products by ultrasound-assisted technology and quality characterisation of the end product. Int. J. Food Sci. Technol. 2017, 52, 2078–2087.

Amoriello, T.; Mellara, F.; Ruggeri, S.; Ciorba, R.; Ceccarelli, D.; Ciccoritti, R. Artichoke By-Products Valorization for Phenols-Enriched Fresh Egg Pasta: A Sustainable Food Design Project. Sustainability 2022, 14, 14778. https://doi.org/10.3390/su142214778

A: We thank the reviewer for comments and suggestion dealing an improvement of the manuscript. All changes made during the revision process are showed using a red font. We added into materials and method section more details concerning the protocol followed by Farmalabor srl to obtain the extract of artichoke leaves. This was included into a separate section, i.e., section 2.2 (LL 122-140). Furthermore, according with the reviewer’s suggestions, we added a short discussion concerning the comparison between our work and the two suggested studies at lines 483-489.

Further changes must be entered.

Equalize the number of significant figures in the means and SDs and in the p-values in 3.1 and 3.1.2, figures 1b and 2.

A: Thank you, both figures have been improved equalizing numbers.

Figure 3 and paragraph 3.2.1: Depicting only two PCs in the PCA when the explained variance is 55% is incorrect. It is suggested to the authors to use at least 3 components, or, better, to tabulate the contributions of the different variables to the loadings. Therefore, the description of the results in paragraph 3.2.1 needs to be rewritten.

A: Thank you. To deeply describe the results deriving from the analysis of principal components we provided a supplementary figure (labeled as Supp. Fig. S2) showing the loading plot of the first three principal components and the variance explained by the first five principal components. To cite this change into the main text, a short note was included at LL 353-354.

Reviewer 2 Report

The subject of the manuscript is interesting and important in the context of searching food with health benefits. Introduction is comprehensive and give relevant background. The presentation of the results is strength of the manuscript.

However, some methodological aspects should be clarified before acceptance for publication and some points need to be corrected.

1)       The aim of the study should be better highlighted. Now it is just description of the investigation.

2)       No information on extract (AE) preparation was provided.  Was it a commercially available product provided by Farmalabor Srl? – it should be clarified

3)       Amount of extract added to bread (proportion) should be added.

 4)      “was characterized by 5% of titratable chlorogenic acid” -  in such method total titratable acids were characterized.

5)       2.3.2. Total phenols (…) – total phenols (TPC) was not assessed

6)       “methanolic extracts (ME) were obtained” – more detail is needed

7)       2.4. Correct the indexes in chemical formulas.

8)       Page 5 „The Gas Chromatography-Mass Spectrometry (GC-MS) generated a chromatogram with peaks representing individual compounds.” – unnecessary statement. This is obvious

9)       2.6. – reedit the title of the section to better reflect the content.

10)   The choice of cell lines for the study should be justified.  Add the explanation to Discussion. Moreover, why different lines were used for particular tests (human keratinocytes were used to assess the protective activity only, and Caco-2 for the other assays)

11)   Lack of experimental data regarding on H2O2 induced stress (concentration, time of incubation, time of incubation with supernatant after stress induction…)

12)   2.6.4. section: Interferon-gamma (IFN-γ) or tumor necrosis factor (TNFα)? And in Figure 6: “interferon-alpha (IFN- α)” – so which factor did Author measure?

13)   after 24h of exposure to lipopolysaccharides” – at what concentration? It is not clear whether LPS was added simultaneously with supernatant or firstly  stress was induced with LPS and after 24h supernatant was added.

14)   3.1.2 „The free radical DPPH was assessed in breads” – reedit. Truly, free radical  scavenging capability/activity was assess

15)   “bread scavenging activity” – not precise. Correct “free radical  scavenging activity”

16)   Figure 5A: H2O2 addition should be marked under X-axis

17)   3.3.2. „Based on TNF-α and IL1-β expression, Caco-2 cells were also used to estimate the pro-inflammatory activity of supernatants” – unclear sentence. Proinflammatory or  anti-inflammatory activity of supernatants?

18)   Discussion: „natural sources of phenol” –  correct: phenols

19)   „Accounting for the presence of chlorogenic acid, the AE increased more than 10-fold the antioxidant activity of the relative breads” - The presence of chlorogenic acid in bread was not confirmed.  It is doubtful that it did not degrade at a high temperature

20)   Figure 6 legend: „Relative quantification (RQ) (…) from E. coli trigger” – „trigger” is unnecessary

Author Response

#R2.

The subject of the manuscript is interesting and important in the context of searching food with health benefits. Introduction is comprehensive and give relevant background. The presentation of the results is strength of the manuscript. However, some methodological aspects should be clarified before acceptance for publication and some points need to be corrected.

A: We thank the reviewer for comments and suggestion dealing an improvement of the manuscript. In order to describe changes made to the manuscript, these were listed below as a point by point answering to the related reviewer’s suggestion. Furthermore, all changes made during the revision process are showed using a red font.

1)       The aim of the study should be better highlighted. Now it is just description of the investigation.

A: The aim of the study was better described and included into the introduction section at LL 91-92.

2)       No information on extract (AE) preparation was provided.  Was it a commercially available product provided by Farmalabor Srl? – it should be clarified

A: Thank you for the suggestion. To better explain the use of a commercially available product of Farmalabor srl, a separate section (2.2.) concerning the AE was added into the materials and methods section (LL. 122-140). Moreover, this section also included the description of the protocol provided by Farmalabor srl and that was followed to obtain the AE.

3)       Amount of extract added to bread (proportion) should be added.

A: Thank you for the suggestion, this information was provided in section 2.3 at LL. 151-152.

 4)      “was characterized by 5% of titratable chlorogenic acid” -  in such method total titratable acids were characterized.

A: Thank you. As previously mentioned, the detail concerning the spectrophotometric-based estimation of chlorogenic acid was included in section 2.2.

5)       2.3.2. Total phenols (…) – total phenols (TPC) was not assessed

A: Thank you for your note. The section title was changed.

6)       “methanolic extracts (ME) were obtained” – more detail is needed

A: Thank you for your suggestion. The part of the method was added in section 2.4.2 at lines 171-174.

7)       2.4. Correct the indexes in chemical formulas.

A: Thank you for your suggestion. All indices were modified to the correct chemical form at lines 195-197.

8)       Page 5 „The Gas Chromatography-Mass Spectrometry (GC-MS) generated a chromatogram with peaks representing individual compounds.” – unnecessary statement. This is obvious.

A: Thank you for your suggestion. This note was deleted.

9)       2.6. – reedit the title of the section to better reflect the content.

A: Thank you for your suggestion. A different title of the section (now labeled as 2.7) was provided at line 238.

10)   The choice of cell lines for the study should be justified. Add the explanation to Discussion. Moreover, why different lines were used for particular tests (human keratinocytes were used to assess the protective activity only, and Caco-2 for the other assays).

A: Thank you for your suggestion. The choice of using two different cell lines took into account the previous validate protocols. Moreover, while for the cytokines assay Caco-2 cell line is one of the most used, a previous work suggested as Caco-2 cells are not suitable for the oxidative stress assay. Both these considerations are now included in the improved version of the manuscript at LL 489-498 also including appropriate references for the discussion.

11)   Lack of experimental data regarding on H2O2 induced stress (concentration, time of incubation, time of incubation with supernatant after stress induction…)

A: Thank you for your suggestion, more details concerning the oxidative stress experiments have been included at lines 276-281.

12)   2.6.4. section: Interferon-gamma (IFN-γ) or tumor necrosis factor (TNFα)? And in Figure 6: “interferon-alpha (IFN- α)” – so which factor did Author measure?

A: We apologize for the mistake. The tumor necrosis factor-alpha (TNF-α) expression was determined in our experimental setting and not the Interferon-gamma (IFN-γ). This was checked in the text of the manuscript.

13)   „after 24h of exposure to lipopolysaccharides” – at what concentration? It is not clear whether LPS was added simultaneously with supernatant or firstly  stress was induced with LPS and after 24h supernatant was added.

A: Thank you for your suggestion, also in this case, more details concerning the LPS-based experiments have been included in section 2.7.4 in order to clarify the amount of LPS used to induce pro-inflammatory expression as well as the detail about the simultaneous presence of LPS and supernatants in experimental theses.

14)   3.1.2 „The free radical DPPH was assessed in breads” – reedit. Truly, free radical  scavenging capability/activity was assess.

A: Thank you, this was revised. L 332-333.

15)   “bread scavenging activity” – not precise. Correct “free radical  scavenging activity”.

A: Thank you for your suggestion, the sentence was changed. L 337.

16)   Figure 5A: H2O2 addition should be marked under X-axis.

A: Thank you for your suggestion, the figure was improved and added to the main text.

17)   3.3.2. „Based on TNF-α and IL1-β expression, Caco-2 cells were also used to estimate the pro-inflammatory activity of supernatants” – unclear sentence. Proinflammatory or  anti-inflammatory activity of supernatants?

A: Thank you, this was a refuse. The term was changed. L 448.

18)   Discussion: „natural sources of phenol” –  correct: phenols.

A: Thank you for your suggestion, it was done. L 473.

19)   „Accounting for the presence of chlorogenic acid, the AE increased more than 10-fold the antioxidant activity of the relative breads” - The presence of chlorogenic acid in bread was not confirmed.  It is doubtful that it did not degrade at a high temperature.

A: Thank you for the suggestion, the sentence was revised to avoid an incorrect meaning.

20)   Figure 6 legend: „Relative quantification (RQ) (…) from E. coli trigger” – „trigger” is unnecessary.

A: Thank you for the suggestion, this was deleted.

Round 2

Reviewer 1 Report

The authors improved the manuscript by accepting reviewers' suggestions

Reviewer 2 Report

All corrections have been made and manuscript has been improved.

I have only two minor suggestions:

dicafeylquinic acids  - correct to „dicaffeoylquinic acid”

line 277: lack of space „400μM”